# Prediction of the Ideal Implant Size Using 3-Dimensional Healthy Breast Volume in Unilateral Direct-to-Implant Breast Reconstruction

**DOI:** 10.3390/medicina56100498

**Published:** 2020-09-24

**Authors:** Jeong-Hoon Kim, Jin-Woo Park, Kyong-Je Woo

**Affiliations:** Department of Plastic and Reconstructive Surgery, Ewha Womans University Mokdong Hospital, College of Medicine, Ewha Womans University, 1071 Anyangcheon-ro, Yangcheon-gu, Seoul 07985, Korea; kimsbrothers@hanmail.net (J.-H.K.); burnscar@naver.com (J.-W.P.)

**Keywords:** 3D breast volume, direct-to-implant breast reconstruction, estimation of ideal implant size

## Abstract

*Background and objectives:* There is no consensus regarding accurate methods for assessing the size of the implant required for achieving symmetry in direct-to-implant (DTI) breast reconstruction. The purpose of this study was to determine whether the ideal implant size could be estimated using 3D breast volume or mastectomy specimen weight, and to compare prediction performances between the two variables. *Materials and Methods*: Patients who underwent immediate DTI breast reconstruction from August 2017 to April 2020 were included in this study. Breast volumes were measured using 3D surface imaging preoperatively and at postoperative three months. Ideal implant size was calculated by correcting the used implant volume by the observed postoperative asymmetry in 3D surface imaging. Prediction models using mastectomy weight or 3D volume were made to predict the ideal implant volume. The prediction performance was compared between the models. *Results*: A total of 56 patients were included in the analysis. In correlation analysis, the volume of the implant used was significantly correlated with the mastectomy specimen weight (R2 = 0.810) and the healthy breast volume (R2 = 0.880). The mean ideal implant volume was 278 ± 123 cc. The prediction model was developed using the healthy breast volume: Implant volume (cc) = healthy breast volume × 0.78 + 26 cc (R2 = 0.900). The prediction model for the ideal implant size using the 3D volume showed better prediction performance than that of using the mastectomy specimen weight (R2 = 0.900 vs 0.759, *p* < 0.001). *Conclusions*: The 3D volume of the healthy breast is a more reliable predictor than mastectomy specimen weight to estimate the ideal implant size. The estimation formula obtained in this study may assist in the selection of the ideal implant size in unilateral DTI breast reconstruction.

## 1. Introduction

Nipple sparing mastectomy (NSM) and direct-to-implant (DTI) breast reconstruction have been gaining popularity because it is oncologically safe, requires less surgery and fewer visits, and is more cost-effective than two-stage expander/implant reconstruction [1,2,3]. Surgeons choose the implants considering the weight of the resected breast tissue, the patient anatomy as assessed by subjective linear measurements, the surgeon’s experience, and the availability of implants. However, linear measurements including the height, width, and projection of the breasts are insufficient to describe the breast shape and size accurately, and small measurement discrepancies may lead to variations in volumetric implant size estimation and potentially unacceptable asymmetry. Although various methods have been developed, there is no universally accepted standard method for determining breast volume and for choosing the optimal implant size for breast reconstruction.

In the last decade, advances in 3-dimensional surface imaging have produced techniques for handling vast data formats efficiently and generating precise 3-dimensional surface images [4,5,6]. The breast volume can be measured preoperatively using 3D surface imaging, and the 3D volumes are known to be significantly correlated with the mastectomy specimen weight [7]. Similar to the mastectomy specimen weight, the preoperative 3D volume of the breasts can assist the surgeon in calculating the ideal volume of the implant. However, there have been no guidelines developed regarding how to estimate ideal implant volume using 3D volume data of the breast. The purpose of this study was to determine whether the ideal implant size could be estimated using 3-dimensional breast volume or mastectomy specimen weight, and to compare prediction performances between these two variables.

## 2. Materials and Methods

### 2.1. Study Population and Data Collection

The study was conducted in accordance with the Declaration of Helsinki, and the protocol was approved by the institutional review board of Ewha Womans University Mokdong Hospital (No. 2020-07-046). Prospectively recorded data from consecutive patients who underwent immediate unilateral DTI reconstruction at a single institution between August 2017 and April 2020 were retrospectively reviewed. Patients who underwent bilateral breast reconstruction, simultaneous contralateral augmentation/reduction surgery, previous surgery on the affected breast side, and patients with incomplete records were excluded. Data on patient demographics, surgical procedures, mastectomy specimen weight, and implant size used were collected. Breast volumes were measured using 3D surface imaging preoperatively and at postoperative three months. The primary outcome was the ideal implant size, which was calculated by correcting the used implant volume by the observed postoperative asymmetry in 3D surface imaging. The independent variables were either the mastectomy specimen weight or the preoperative 3D volume of the healthy breast.

### 2.2. Method for 3-Dimensional Surface Imaging and Volume Extraction

Volumetric assessment of the breast was obtained using a Crisalix 3D^®^ imaging scan (Crisalix, Lausanne, Switzerland). The Crisalix system is a cloud-based, 3-dimensional simulation program. 3D surface scanning was performed using a 3D sensor attached to a portable tablet to scan the patient’s front and both sides in a standing position. The total time for each individual scan was a few seconds, and the total procedure time, including marking of breast landmarks, takes less than 10 s. All 3D scans were performed with the patient in the same position. Patients were scanned in a standing position with the back of the feet and the shoulders touching the wall, and arms abducted with both wrists placed on the hips of the patients.

Data from the 3-dimensional scan are uploaded and merged to generate a 3-dimensional surface image, which is then rendered. This software program can then generate the curvature of the simulated chest wall and the indicated breast boundary from the torso curved to match the real body shape and thereby calculate the volume of the 3D breast image [8]. After completing the 3-dimensional surface image, the volume of the breast soft tissue was measured with consideration of 3 mm skin thickness (Figure 1). Postoperatively, the 3D surface imaging was repeated to compare the volumes of the reconstructed and healthy breasts at three months postoperative follow-up using the same protocol.

### 2.3. Reconstruction after Nipple-Sparing Mastectomy

Oncologic surgeons performed the mastectomy, and the senior author (K.J.W.) performed all reconstructions. The implant size and type were determined by the surgeon considering the patient’s breast dimensions, contralateral breast volume measured before surgery, and the resected mastectomy specimen weight to achieve symmetric breasts. The size of the implant used was determined after two or three temporary sizers were inserted and checking the breast symmetry by visual inspection and palpation. The implants were placed in the subpectoral or prepectoral spaces. In subpectoral DTI, an acellular dermal matrix (human cadaveric) of 6–8 × 16–18 cm size was used for inferior and lateral support and implant coverage. In prepectoral DTI, a 16 × 16 cm or 18 × 18 cm acellular dermal matrix was used to cover the implant.

### 2.4. Calculation of Ideal Implant Size

Because the implants used were not always the ideal size for symmetry in unilateral DTI breast reconstructions, an ideal implant size was calculated by correcting the used implant volume by the observed postoperative asymmetry in 3D surface imaging. The ideal implant volume = Inserted implant volume − β × (Surgical side breast volume − Contralateral side breast volume measured postoperatively at three months after surgery). The β was the slope of the linear regression model that was obtained using the preoperative breast volume as an independent variable and the inserted implant volume as the dependent variable. If the β was 0.7, a 100 cc increase of the breast volume resulted in a 70 cc increase of the implant volume in the linear regression model.

### 2.5. Statistical Analysis

The mean with standard deviation or median with interquartile range were used to summarize continuous variables based on the distribution of the data. Pearson correlation coefficients between implant size and morphological factors, including mastectomy specimen weight and healthy breast volume, were first examined to determine the most suitable references for implant size choice. Linear regression analysis was performed to develop formulas predicting the optimal inserted implant volume.

After calculation of the ideal implant sizes, linear regression models were used to develop formulas to predict the ideal implant volume for symmetry using mastectomy specimen weight and preoperative healthy breast volume as predictor variables. The prediction performances using the two predictor variables were compared. Residual analysis was performed to assess the appropriateness of the linear regression model. The statistical significance was determined by *p* < 0.05. All analyses were performed using SPSS version 23.0 (SPSS Inc., Chicago, IL, USA).

## 3. Results

### 3.1. Patient Demographics and Operative Data

A total of 56 patients undergoing immediate unilateral DTI reconstruction were included in the analysis. The patients’ mean age was 47.95 ± 8.44 years (IQR, 43.5–52.3) with a mean BMI of 22.77 ± 2.50 kg/m^2^ (IQR, 20.92–24.25). Nipple-sparing mastectomy was performed in 85.7% (48 of 56 patients), and skin-sparing mastectomy was performed in the remaining cases. Prepectoral placement of the implant was performed in 46.4%, and subpectoral placement was performed in 53.6%. The mean preoperative volume of the affected breast was 318 ± 154 cc (IQR, 194–408) and that of the contralateral unaffected breast was 323 ± 150 cc (IQR, 203–380). The mean mastectomy specimen weight was 287 ± 128 g (IQR, 181–348) and the mean volume of inserted implant was 288 ± 107 cc (IQR, 200–375) (Table 1).

In the analysis of preoperative 3D volume of the healthy and affected breasts, mean volume differences of the breasts were 49.5 ± 39.8 cc (IQR, 23.0–76.0 cc) (Figure 2). The mean percentage of volume differences was 15.8 ± 13.4% (IQR, 8.5–23.1%), and 32.1% of the patients (18 of 56) had over 20% volume differences between the healthy and affected breasts.

### 3.2. Prediction Model for the Inserted Implant Volume

The Pearson correlation coefficient of the mastectomy specimen weight was 0.900 (*p* < 0.001). In the linear regression analysis, a prediction model was developed (Figure 3).

Inserted implant volume = 0.75 × mastectomy specimen weight (g) + 72 cc (R2 = 81.0%, *p* < 0.001).

The Pearson correlation coefficient of the healthy breast volume was 0.938 (*p* < 0.001). In the linear regression analysis, the inserted implant volume was better estimated with the model using healthy breast volume.

2.Inserted implant volume = 0.66 × healthy breast volume + 71 cc (R2 = 88.0%, *p* < 0.001).

The results of the residual analysis satisfied the assumptions of the linear regression models.

### 3.3. Prediction Model for Ideal Implant Volume

Because the inserted implant volume could not be considered as an ideal implant volume for symmetry, ideal implant volume was calculated by comparing the postoperative volumes of both breasts. The mean volume of the ideal implant size was 278 ± 123 cc (IQR, 181–338 cc). The prediction model of an ideal implant volume was as follows (Figure 4).

Mastectomy specimen weight as a predictor variable.

Ideal implant volume = mastectomy specimen weight × 0.84 + 37 cc (R2 = 75.9%, *p* < 0.001).

2.Healthy breast volume as a predictor variable.

Ideal implant volume = healthy breast volume × 0.78 + 25 cc (R2 = 90.0%, *p* < 0.001).

The results of the linear regression models showed that an ideal implant volume could be predicted by both healthy breast volume and mastectomy specimen weight. In terms of prediction performance, the ideal implant volume could be better estimated using the 3D volume of the healthy breast compared to the mastectomy specimen weight (coefficient of determination = 90.0% vs. 75.9%). The results of the residual analysis satisfied the assumptions of linear regression models.

## 4. Discussion

The selection of an ideal size of implant is crucial for achieving symmetry in immediate DTI breast reconstruction. The current study demonstrated that the ideal implant size could be estimated by a linear regression model using either mastectomy specimen weight (*p* < 0.001) or the 3D volume of the healthy breast (*p* < 0.001). Moreover, we found that the 3D volume could predict the ideal implant size more accurately than the mastectomy specimen weight (R2 = 0.900 vs. 0.759).

Many surgeons subjectively evaluate the breast volumes and symmetry. However, subjective measurement of breast volume cannot be reliable and is not accurate [9]. Mastectomy specimen weight has been proposed to be used as a reference for the selection of ideal implant size [10,11,12,13]. The density of mastectomy specimen weight is known to be 1.06 g/mL [14]. However, the density of the breast is different among patients because the proportions of fibro-glandular tissue and fat tissue volume of the breast are different. The tumor itself can also change the density and weight of the affected breast.

Furthermore, innate breast asymmetry cannot be taken into account when using the mastectomy specimen as a reference to estimate the ideal implant size. Recently, Liu et al. found that the incidence of significant asymmetry of the breast mound was 94 percent [15]. We also found that most patients had innate asymmetry of their breast volumes. The mean percentage of volume differences was 15.8 ± 13.4% (IQR, 8.5–23.1%), and about one-third of the patients (18 of 56) had over 20% volume differences between the healthy and affected breasts in our study. Georgiou et al. reported that the implant size could be predicted by the mastectomy specimen weight using a linear regression model, but there were limitations since the coefficient of determination (R2) was less that 0.5 (R = 0.66) [11].

2-dimensional images of CT or MRI and subsequent 3D reconstruction can be used for breast volume measurement [12,16,17,18,19,20]. However, 3D reconstruction of the 2D images are complex and need additional software. Moreover, it is not completely objective because boundary annotation of the breast tissue has to be performed manually or threshold of breast tissue has to be arbitrarily determined on the 2D images.

Recently, 3D surface imaging has gained acceptance in clinical use in breast surgery [4,5,6]. 3D surface imaging has been used for the selection of implant size and for creating simulations for augmentation mammoplasty [21,22,23,24]. Yip et al. reported that breast volume could be measured by 3D surface imaging (Pearson’s correlation, R = 0.95, *p* < 0.001) [7]. Roostaeian et al. evaluated the accuracy of the 3D surface imaging and reported that preoperative simulation by 3D surface imaging can predict postoperative breast volume with more than 90% accuracy [25]. Previous studies have suggested the potential use of 3D surface imaging for measuring the breast volume for selection of the ideal implant size in DTI breast reconstructions. We found that the 3D volume of the healthy breast measured by 3D surface imaging could be used for estimation of the ideal implant size. We also demonstrated that the 3D volume of the healthy breast showed better prediction performance than the mastectomy specimen weight (coefficient of determination, R2 = 0.900 vs. 0.759). Implant size can be easily calculated by multiplying the healthy breast volume by 0.78 and adding 25 cc (ideal implant volume = healthy breast volume × 0.78 + 25 cc).

In order to develop a predictive model for satisfactory results after surgery, it is desirable to set the dependent variable as an ideal implant size rather than the inserted implant size. Most of the previous studies developed a prediction model based on the inserted implant. However, the inserted implant was not validated as an ideal implant size because postoperative evaluation was not performed in the previous studies. In the present study, we calculated the ideal implant size by correcting the inserted implant volume using the observed postoperative asymmetry in postoperative 3D surface imaging. Pohlmann et al. also calculated the ideal implant size by simply adding or subtracting the postoperative volume differences of both breasts [26]. However, 100 cc differences of the breast volume do not correspond to 100 cc differences of an implant. In the analysis of our data using inserted implant size and 3D breast volume, beta was 0.66 (R2 = 88.0%, *p* < 0.001). The results suggested that a 100 cc difference corresponded to 66 cc of implant. Therefore, multiplying beta by the postoperative volume difference would be better for calculation of the ideal implant size.

One of significant findings of our study was that the beta values of the prediction model using 3D breast volume (0.78) and mastectomy specimen weight (0.84) were both less than 1. This result suggests that the ideal implant size should be smaller than the 3D breast volume or mastectomy specimen weight except when the breast size is small (<200 cc) (Figure 5). According to our prediction model, difference between ideal implant volume and 3-D volume of the healthy breast (or mastectomy weight) becomes larger as the breast size increases. This would be true because extent of oncologic resection is usually larger than the dimensions of the implant. In this case, the inserted implant makes central portions of the breast mound without covering peripheral area of the mastectomy defect. Similar with our study, Back et al. reported that long-term patient satisfaction was highest in a patient group whose implant volume to mastectomy specimen weight was 71.9% [10].

Considering that our retrospective data set was relatively small, and the developed formulas have not been fully tested, the generalizability of these two formulas may requires more testing, especially in different populations. However, a relatively high coefficient of determination (R2 = 0.90) could be obtained with statistical significance. The coefficient of determination in our prediction model (R2 = 0.90) was higher than that of previous studies [11,13,26]. In terms of selection of the ideal implant in unilateral DTI breast reconstruction, implant volume is not the only factor to be considered for breast symmetry. Other various factors including breast width, breast height, projection, upper pole fullness, and degree of ptosis need to be considered together for the ideal implant volume for symmetry. Additionally, the size of the acellular dermal matrix influences the final volume of the reconstructed breast. The thickness and dimensions of the acellular dermal matrix should be considered, especially in prepectoral DTI, in which the implant is covered 360 degrees with the acellular dermal matrix.

## 5. Conclusions

Healthy breast volume measured by 3D surface imaging is a more accurate predictor than mastectomy specimen weight to estimate the implant volume for symmetry in DTI breast reconstruction. The estimation formula obtained in this study may assist in the selection of ideal implant size in unilateral DTI breast reconstruction.

## Figures and Tables

**Figure 1 medicina-56-00498-f001:**
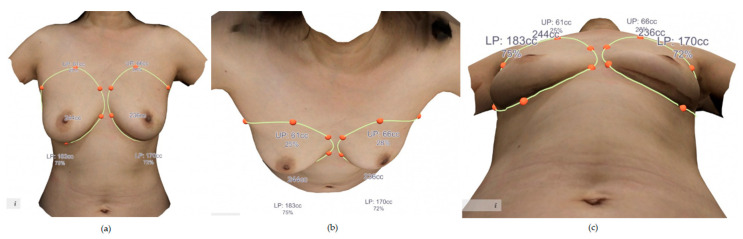
A case example of preoperative 3-D volumes of the breasts. (**a**) Anterior view. (**b**) Cephalic to caudal view. (**c**) Caudal to cephalic view.

**Figure 2 medicina-56-00498-f002:**
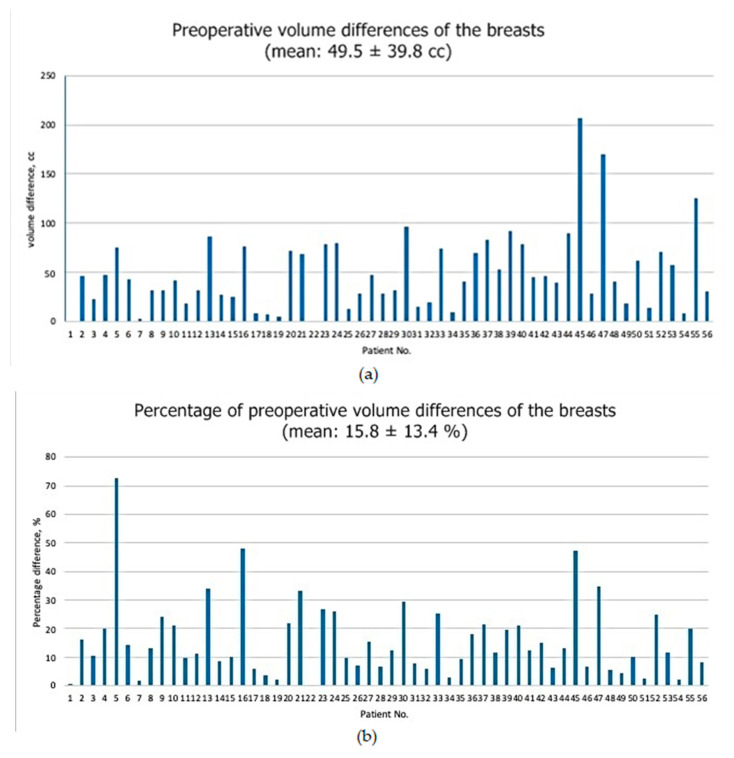
(**a**) Preoperative volume differences of the breasts. (**b**) Percentage of preoperative volume differences of the breasts. The percentage of volume differences were calculated by (1-reconstruction side breast volume/healthy breast volume) × 100.

**Figure 3 medicina-56-00498-f003:**
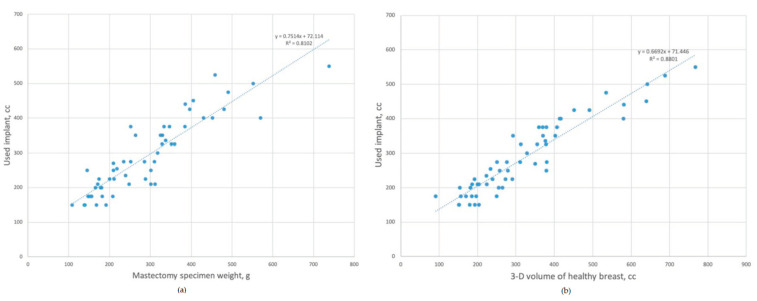
Predication model for inserted implant. (**a**) Scatterplot and the linear regression model using mastectomy specimen weight as a predictor variable. (**b**) Scatterplot and the linear regression model using 3-D volume of healthy breast as a predictor variable.

**Figure 4 medicina-56-00498-f004:**
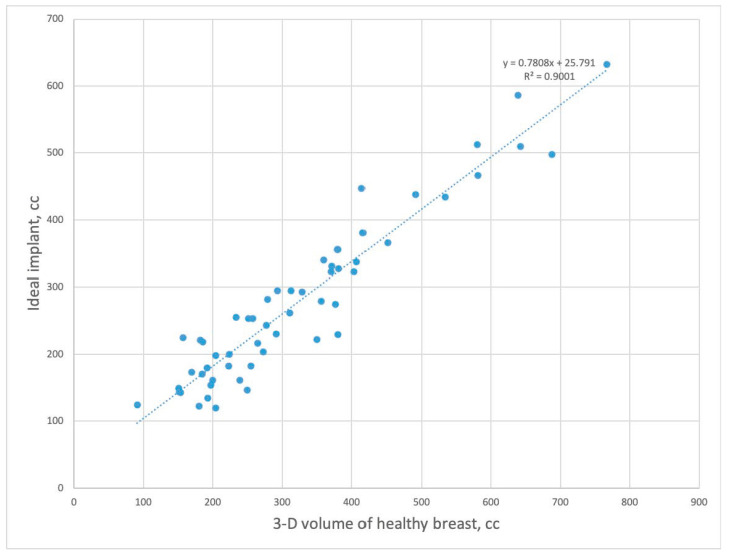
Predication model for ideal implant. Scatterplot and the linear regression model using 3-D volume of healthy breast as a predictor variable.

**Figure 5 medicina-56-00498-f005:**
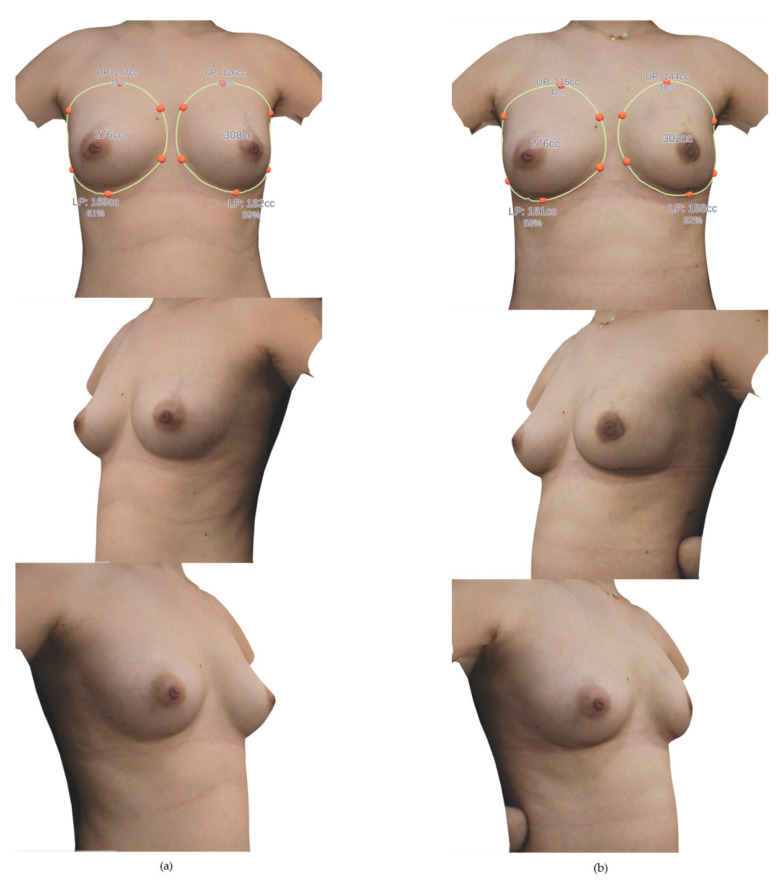
A case example of pre and postoperative 3-D volumes of the breasts. (**a**) Preoperative 3-D image. A 41-year-old woman with a diagnosis of left breast cancer. The volume of both breasts was 276 cc and 308 cc. The resected mastectomy specimen weight was 252 g and the inserted implant size was 275 cc. The operation was performed through periareolar incision, and the implant was inserted into the subpectoral plane. (**b**) Postoperative 3-D image. The volume of both breasts after three months of surgery was 276 cc and 302 cc. The volume on the reconstruction side was 9.4% larger.

**Table 1 medicina-56-00498-t001:** Clinical and surgical characteristics.

No. of patients	56
Age, mean ± SD, yr	47.95 ± 8.44
BMI, mean ± SD, kg/m^2^	22.77 ± 2.50
Cancer laterality	
No. of right (%) No. of left (%)	35 (62.5) 21 (37.5)
Mastectomy type	
No. of nipple-sparing (%) No. of skin-sparing (%)	48 (85.7) 8 (14.3)
Mastectomy specimen weight, mean ± SD, g	287.6 ± 128.2
Inserted implant volume, mean ± SD, cc	287.5 ± 107.0
Inserted ADM size, mean ± SD, cm^2^	204.0 ± 82.9
Preoperative volume of the breasts	
Pre-operative volume of affected breast, mean ± SD, cc Pre-operative volume of contralateral unaffected breast, mean ± SD, cc	317.6 ± 154.3 322.9 ± 150.0
Postoperative volume of the breasts	
Post-operative volume of affected breast, mean ± SD, cc Post-operative volume of contralateral unaffected breast, mean ± SD, cc	336.8 ± 147.8 321.2 ± 161.1

SD, standard deviation; BMI, body mass index; ADM, acellular dermal matrix.

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
