# Peer review of "Prediction of the Ideal Implant Size Using 3-Dimensional Healthy Breast Volume in Unilateral Direct-to-Implant Breast Reconstruction"

_medicina, 2020, doi:10.3390/medicina56100498_

Round 1

Reviewer 1 Report

The authors propose an interesting model that uses 3D surface imaging measurements of healthy breast volume to predict the ideal implant size for unilateral DTI reconstruction. Overall, the paper is clear and well written with only minor grammatical errors that need correcting. The purpose of the study and the need to develop such a model in comparison to current applications is well presented, so too the methods section, particularly the detailed explanation on what variables went into the simple linear regression model. My only concern with this section regards ethics. It is unclear what was approved from the institutional review board or whether patients consented to their data being included in the study – please clarify. The results were clearly presented with the figures/tables complementing the text. I suggest a better resolution image for figure 2 be used and in figure 2 (above) ‘volume’ is misspelt. The discussion articulated the importance of the proposed model, which uses 3D surface imaging as opposed to subjective evaluation by surgeons, mastectomy specimen weight and 2D measurements. Whilst also accepting that this is a predictive model that needs confirmatory testing and that there are other important factors for breast symmetry that will need to be added to this model. Perhaps a future study can look at performing multiple linear regression to see what variables should/should not be included in the model.

You have shown that healthy breast volume measured by 3D surface imaging is statistically a better predictor to estimate the ideal implant size in unilateral DTI reconstruction. I suggest you amend the title of the article as it’s quite long and does not reflect your findings.

Author Response

Dear reviewer

Thank you very much for your quick and attentive review.

Firstly, we modified the title of the manuscript into ‘Prediction of the ideal implant size using 3-dimensional healthy breast volume in unilateral direct-to-implant breast reconstruction’ according to your suggestion. We changed the Figure 2 into better resolution images, and the misspelling has been corrected. 

Secondarily, the current study was retrospective study and the study protocol was approved by the IRB. Informed consent for the potential used of the patients’ photograph was achieved for all patients. We have inserted a sentence “The study was conducted in accordance with the Declaration of Helsinki, and the protocol was approved by the institutional review board of Ewha Womans University Mokdong Hospital (No. 2020-07-046)” to the first paragraph of the “Materials and Methods”.

Finally, based on your opinion, we would like to conduct research through multiple linear regression to find out what variables should/should not be included in the model.

Thank you for your kind consideration of this manuscript.

Sincerely,

Dr. Kyong-Je Woo, M.D., Ph.D.

Department of Plastic and Reconstructive Surgery, Mokdong Hospital, College of Medicine, Ewha Womans University

Anyangcheon-ro 1071, Yangcheon-gu

Seoul, Republic of Korea

[email protected]

Reviewer 2 Report

I have carefully reviewed the manuscript submitted by J.H. Kim and al. The article highlights the importance and the efficiency of 3-dimensional breast volume and mastectomy specimen weight in one stage direct-to-implant breast reconstruction. Generally speaking, this is a well-written manuscript. The topic of the current study is interesting. This estimation formula definitely can improve the quality of the unilateral breast reconstruction, helping both the plastic surgeon and the patient. The background of the study, study design, data presentation, but also the results are good.  Moreover the article shows good English and the bibliography is coherent and recent.

Author Response

Dear reviewer

Thank you very much for your kind and attentive review.

We agree with your opinion that estimation formula can improve the quality of the unilateral breast reconstruction, helping both the plastic surgeon and the patient.

Sincerely,

Dr. Kyong-Je Woo, M.D., Ph.D.

Department of Plastic and Reconstructive Surgery, Mokdong Hospital, College of Medicine, Ewha Womans University

Anyangcheon-ro 1071, Yangcheon-gu

Seoul, Republic of Korea

[email protected]

Reviewer 3 Report

I read with great interest this manuscript on estimating the ideal implant size using 3-dimensional breast volume. The authors proposed a method that could be performed preoperatively and take less than 1 minute. The methods and results sections are well done. Their discussion also included important considerations about their study limitations. The manuscript could benefit from the following:

-Discuss the availability of software for 3D analysis.

-How could 3D smartphone cameras potentially impact the field? 

Author Response

Dear reviewer

Thank you for your kind consideration of this manuscript.

Currently widely used 3D imaging programs include Vectra 3D® (Canfield Scientific, Parsippany, New Jersey, USA), Crisalix 3D® (Crisalix, Lausanne, Switzerland), and so on. They all reconstruct a real human body in 3D using vast cloud-based data. They are being used during private, in-office consultations with patients who are considering various plastic surgeries and also helps patients to maintain realistic expectations and boosts their confidence. Because of issue of conflict of interest this content was not added in the manuscript.

We performed 3D imaging over a short period of about 10 seconds and reconstructed it based on cloud before surgery. Through this process, we have developed a model that can more accurately predict the size of the ideal implant. A smartphone 3-D camera will be popularized in the near future. As a result, we are confident that the establishment of an individualized surgery plan and patient consultation using 3D reconstructive images using a smartphone will be much easier and common in the future.

Sincerely,

Dr. Kyong-Je Woo, M.D., Ph.D.

Department of Plastic and Reconstructive Surgery, Mokdong Hospital, College of Medicine, Ewha Womans University

Anyangcheon-ro 1071, Yangcheon-gu

Seoul, Republic of Korea

[email protected]
